# Diagnostics of Exercise-Induced Laryngeal Obstruction Using Machine Learning: A Narrative Review

Rune Mæstad [1,*,†], Haakon Kristian Kvidaland [2,3], Hege Clemm [3,4], Ola Drange Røksund [2,3,5] and Reza Arghandeh [1]

1   Faculty of Engineering and Science, Western Norway University of Applied Sciences, 5063 Bergen, Vestland, Norway
2   Faculty of Health and Social Sciences, Western Norway University of Applied Sciences, 5063 Bergen, Vestland, Norway
3   Department of Pediatric and Adolescent Medicine, Haukeland University Hospital, 5009 Bergen, Vestland, Norway
4   Department of Clinical Science, University of Bergen, 5007 Bergen, Vestland, Norway
5   Department of Head and Neck, Haukeland University Hospital, 5009 Bergen, Vestland, Norway
*   Correspondence: rune.maestad@gmail.com
†   Current Address: Department of Computer Science, Electrical Engineering and Mathematical Sciences, Western Norway University of Applied Sciences, Inndalsveien 28, 5063 Bergen, Vestland, Norway.

**Abstract:** *Objective:* This paper explores machine learning methods for exercise-induced laryngeal obstruction (EILO) diagnostics. Traditional diagnostic approaches like CLE scoring face subjectivity, limiting precise objective assessments. Machine learning is introduced as a theoretical solution to potentially overcome these limitations and improve diagnostic precision. *Methods:* A narrative review was conducted to explore the integration of machine learning techniques in the diagnostics of EILO. *Result:* Three machine learning methods for the segmentation of laryngeal images were discovered: fully convolutional network, Mask R-CNN, and 3D VOSNet. Our findings reveal that the integration of machine learning with EILO diagnostics remains a largely untapped research domain, providing significant room for further exploration. *Conclusions:* The integration of ML techniques for EILO diagnostics has the potential to be a helpful tool for clinicians. The application of computer vision ML methods, such as image segmentation, to delineate laryngeal structures paves the way for a more objective assessment. While challenges persist, especially in differences in patients' laryngeal anatomy, the synergy of ML and medical expertise is an important field to explore in the years to come.

**Keywords:** exercise-induced laryngeal obstruction; CLE-test; machine learning; artificial intelligence; image segmentation

## 1. Introduction

Exercise-induced laryngeal obstruction (EILO) presents a challenging condition marked by the narrowing of laryngeal structures during physical activity, causing significant breathing difficulties [1,2]. Often misdiagnosed as asthma, EILO affects exercise performance and quality of life, with a prevalence ranging from 5% to 8% in the general adolescent population and even more prevalent among athletes and active youth [3–5].

The continuous laryngoscopy exercise test (CLE-test) has been fundamental for establishing the EILO diagnostics and has emerged as the gold standard within the field [6,7]. The test utilizes a fibreoptic laryngoscope, enabling real-time visualization of the larynx during treadmill exercise. For over a decade, the CLE score system developed by Maat et al. [8] has been the preferred method for assessing the severity of EILO in patients. The CLE score is deduced from a clinical visualization of the larynx, as the authors concluded that the vocal fold and supraglottic obstruction are crucial for assessing EILO severity.

These manual methods have been found to have their limitations, with the subjectivity of the CLE score being a notable example [9]. Advancements in machine learning (ML), specifically computer vision, have the potential to address the problem of objectivity. ML algorithms can observe inappropriate patterns in laryngeal movements and serve as a tool for clinicians, assisting in the process of EILO diagnostics. It is important to notice that visualization of the larynx, indicating laryngeal abduction and area for airflow, is often not enough to assess the severity of EILO. Patient's requirements for airflow differ, resulting in variability in the necessary airway space from one patient to another.

This paper makes several contributions to the field of EILO diagnostics using ML methods. First, we provide an overview of state-of-the-art methods for EILO diagnostics, categorizing them into classic, non-machine learning approaches and ML techniques. Second, we delve into the challenges posed by existing manual methods, emphasizing their inherent subjectivity. Furthermore, we explore the application of image segmentation, an ML technique, for the analysis of laryngeal structures within the context of EILO. Importantly, we have not identified any existing review papers specifically tailored to ML-based larynx analysis for EILO diagnostics. Therefore, this review paper aims to fill this research gap by presenting a comprehensive overview of high-performing methods and their architectures, thereby providing valuable insights for future research in the field.

## 2. Overview of Continuous Laryngoscopy Exercise Test

When affected by EILO, the obstruction of the larynx increases with rising levels of exercise. For evaluating the severity, the obstruction of both glottic and supraglottic structures is important [7]. Specifically, the glottic region, including the vocal folds, may abduct, while the supraglottic structures can curve inward during inspiration. These structural changes may cause respiratory challenges during physical activity.

The CLE-test serves as the existing benchmark for diagnosing EILO and allows for laryngeal visualization during treadmill or ergometry-based exercise [10]. In its original configuration [6], the test incorporates a laryngoscope equipped with video recording features inserted through the nose and fixated with a nose clip. The laryngoscope is attached to a customized headset, which also serves to stabilize and reduce movement during the test (Figure 1). However, recording errors like the camera shaking might occasionally occur, resulting in the vocal folds being obscured by surrounding tissues. During the procedure, the patient runs on a treadmill until they experience their most severe symptoms of EILO, or until exhaustion. This enables medical professionals to conduct a comprehensive evaluation of laryngeal function. In certain instances, a cardiopulmonary exercise test (CPET) is integrated into the procedure and collects data like oxygen uptake (VO2), respiratory exchange ratio (RER), and minute ventilation (VE). In addition to these measurements, exercise flow volume loops (EFVLs) are captured to provide insights into respiratory function during exercise. Concurrently, electrocardiography (ECG) can be used to monitor cardiac activity.

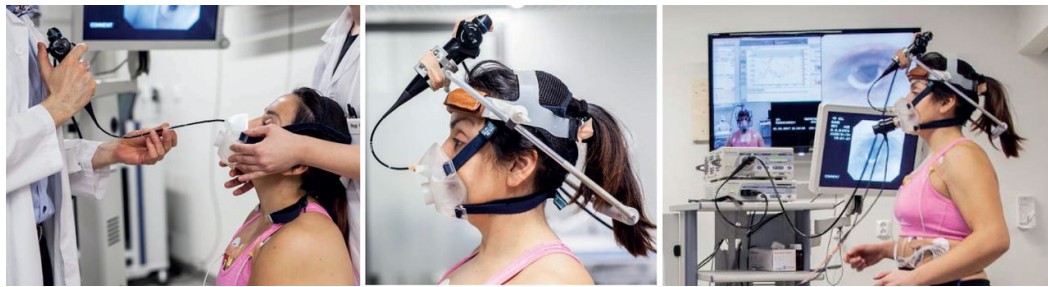

**Figure 1.** Continuous laryngoscopy exercise test employs a laryngoscope attached to a headset along with a facemask. Real-time images of the larynx are displayed on the screen during exercise. [11]. Reproduced with permission of the © ERS 2024: *Eur. Respir. J.* **2017**, *50*, 1602221; DOI: 10.1183/13993003.02221-2016 Published 9 September 2017.

## 3. State-of-the-Art Methods

The objective of this paper is to offer a narrative review centered on the application of ML techniques in the diagnostics of EILO, with a particular focus on laryngeal image segmentation. We conducted searches on academic databases including PubMed and ScienceDirect utilizing key search terms such as "EILO", "larynx", "image segmentation", "semantic segmentation", "machine learning", and "deep learning". Additionally, we examined articles citing Lin et al.'s work from 2017 [12], which served as an inspiration for this review. The search was carried out over the months from September to November 2023. To find state-of-the-art machine learning methods, we narrowed our review to studies published in 2017 or more recent years. Historically, conventional techniques have set the standard for evaluating CLE-test images and measuring the grade of EILO. Nevertheless, recent studies have started using ML methods to analyse and understand larynx movements. The next sections provide a comprehensive exploration of both of these methodologies.

### 3.1. Non-Machine Learning Methods

The CLE-test offers direct visualization of the larynx as exercise progresses, fulfilling the requirement for an endoscopic examination to accurately identify and grade the underlying causes of exercise-induced inspiratory symptoms. Maat et al. [8] suggested a scoring system for laryngeal obstruction based on insights from evaluations on CLE-test videos. This study noted a significant correlation between respiratory distress and laryngeal adduction during exercise, emphasizing that laryngeal obstruction plays a crucial role in exercise-induced inspiratory symptoms.

The CLE scoring system comprises two sub-scores, one for glottic and one for supraglottic movements (Figure 2). The scores reflect the abduction of the vocal folds and supraglottic during inspiration. Both scores range between 0 and 3. A score of 0–1 suggests that glottic and supraglottic levels are considered normal. Scores 2 and 3 increase the probability of EILO, even though some patients with a score of 2 show no respiratory problems. Patients with glottic and supraglottic scores of 1 have been diagnosed with EILO, highlighting the importance of considering factors beyond the CLE score for EILO assessments. The scoring system's subjectivity can lead to inconsistencies and bias in diagnostic outcomes among different clinicians [9]. Nevertheless, this measure is commonly employed in clinics globally.

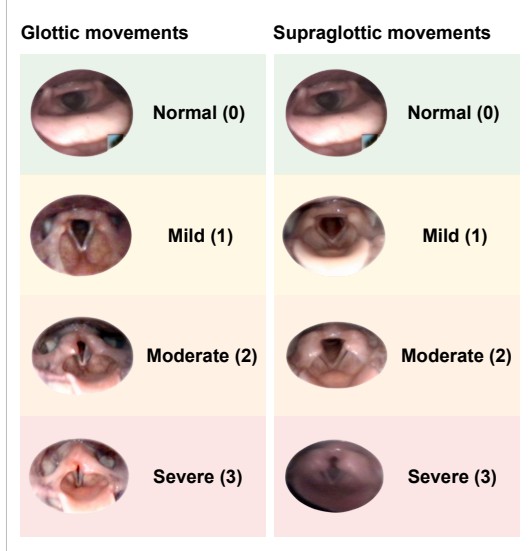

**Figure 2.** CLE score: Scoring system for EILO severity based on movements of laryngeal structures during exercise [13].

Christensen et al. [14] introduced the EILO diagnostic software measuring tool (EILOMEA) for objectively analysing images from the CLE-test. They evaluated its reproducibility and clinical utility in a randomly selected group of individuals. The optimal larynx image during maximal inspiration and expiration within the last 20 seconds of a physical stress test was chosen and imported into EILOMEA. The vocal folds, the larynx's midline, and other essential anatomical points were marked. EILOMEA used these markers to compute cross-sectional areas and distances. Although the EILOMEA software tool calculates using pixel units, the results given to the examiner are relative values, adjusting for potential differences in the laryngoscope's distance from the larynx to ensure uniform comparisons. The software derives three continuous variables from these markings: (1) the ratio of the actual to the maximal lumen at the arytenoid region, indicating the severity of supraglottic obstruction (SO); (2) the ratio of the area between the vocal folds and the distance from the back to the front of the glottis, which indicates the severity of glottic obstruction (GO), known as the EILOMEA glottic obstruction measure 1; and (3) the angle at the vocal folds' anterior commissure that indicates the severity of glottic obstruction, termed the EILOMEA glottic obstruction measure.

Norlander et al. [15] compared the CLE score and EILOMEA methods. They showed that both methods are aligned in their findings. The CLE score method is quicker, allowing for real-time assessment during video playback. In contrast, the EILOMEA method involves multiple steps post-recording and is based on a chosen still image. The CLE scoring uses a four-tier ordinal scale, while EILOMEA employs continuous scales. This makes EILOMEA more suitable for detailed post-intervention tracking and advanced statistical analysis. Consequently, Norlander et al.'s study concludes that EILOMEA might be better suited for research, whereas the CLE score method is likely more practical for routine clinical use.

*3.2. Machine Learning Methods*

In Lin et al.'s study [12], convolutional networks were used to create a deep learning model for quantification and analysis of the laryngeal closure. A convolutional neural network (CNN) was utilized to detect the region of interest (RoI), and a fully convolutional network (FCN) was used to detect objects. The traditional FCN model, as described in this study [16], has a couple of notable shortcomings: (1) Its processing speed is not optimal for real-time analysis and does not effectively capture global context. (2) The FCN's output feature maps are downsampled due to alternating between convolution and pooling layers, which could lead to lower-resolution predictions, causing blurred object boundaries.

A study from Choi et al. [17] implemented a configured instance of the Mask Region-based Convolution Neural Network (Mask R-CNN) architecture [18]. The implementation aimed to segment anatomical structures captured from laryngoscopy videos, including the epiglottis, vocal folds, tongue, and corniculate cartilage. The study resulted in multiple models applicable for image segmentation during endotracheal intubation (ETI), including the Configured Mask R-CNN in addition to DeepLabv3+ and U-Net with EfficientNet-B5 as the encoder. All models performed well, with the Mask R-CNN achieving the highest number of segmented frames per second.

Chen et al. [19] proposed a deep learning architecture, 3D VOSNet, to concentrate on the segmentation of the larynx, focusing on assessing muscle movement in the larynx to diagnose laryngeal-related diseases. Frequently, laryngeal-related diseases are evaluated using laryngeal electromyography (EMG) [20], a method that is generally not well tolerated by patients. Moreover, laryngeal endoscopy is commonly employed to monitor vocal fold movement and assess the extent of glottic closure. Due to the human eye's limited capacity for dynamic object recognition, 3D VOSNet has been developed to segment images to better visualize the larynx. Compared to the others, the advantage of this method is that the input consists of a sequence of images. This incorporates temporal information into the model, enhancing its ability to understand the progression of movement throughout the entire video.

As the realm of ML techniques for image segmentation broadens, it is important to consider additional established methods that may offer significant potential in tackling similar challenges, beyond what we have previously discussed. Residual network (ResNet) [21] is a well-known method mostly used for computer vision tasks. ResNet utilizes residual blocks with skip-connections, contributing to a quicker convergence of the learning process and decreasing training time. Skip-connections have notably impacted advancements in biomedical image segmentation [22]. After its introduction in 2017 [23], transformers have been the state-of-the-art method for natural language processing. Motivated by this scaling success, Dosovitskiy et al. [24] experimented with applying a standard transformer directly to images. Transformers have been applied for medical image segmentation with great results [25].

## 4. Overview of Selected Machine Learning Methods

In this section, we provide an overview of three selected ML methods introduced in the previous section (summarized in Table 1). As we lack access to the datasets utilized for training, the aim is to present the architecture of these methods rather than compare their performance. The selected models are carefully selected based on the following criteria: (1) An image segmentation model with a successful application in segmenting laryngeal images. (2) Preferably, the model application should be related to EILO diagnostics. (3) The model should also be easy to implement or have "ready-to-use" implementations in frameworks like PyTorch or TensorFlow.

**Table 1.** Summary and comparison of the selected segmentation models.

| Model | Results | Evaluation Metrics | Dataset |
|---|---|---|---|
| Fully Convolutional Network [12] | Quantification and analysis of laryngeal movements to aid EILO diagnosis. | IoU [1]: Glottic opening 0.85, vocal fold (left) 0.72, vocal fold (right) 0.65, supraglottis (left) 0.77, supraglottis (right) 0.75 | 806 images extracted from CLE-test data acquired from 194 patients with diverse hardware configurations. Not publicly available. |
| Mask R-CNN [18] | Segmentation of anatomical structures from laryngoscopy images. Used for ETI [2]. | DSC [3]: Tounge 0.12, epiglottis 0.77, vocal folds 0.72, corniculate cartilage 0.57 | 8973 images extracted from 54 cases of intubation videos from clinical emergencies. Data are available on request. |
| 3D VOSNet [19] | Segmentation of endoscopic images of the larynx for more objective diagnostics of laryngeal diseases. | Accuracy: Vocal fold (left) 0.93, vocal fold (right) 0.95, glottic opening 0.90 | 50 laryngoscope videos, each lasting about 10 s, captured at 30 frames per second. A total of 15.000 frames before augmentation. Data are available on request. |

[1] Intersection over Union. [2] Endotracheal intubation. [3] Dice similarity coefficient.

### 4.1. Convolutional Networks

In 2017, Lin et al. [12] proposed a method for quantifying and analysing laryngeal videos. At that time, there were no objective methods to measure and characterize laryngeal obstruction. A primary motivation behind the project was the unreliability of subjective quantification, which could lead to diagnostic errors. The suggested result reviews laryngoscopic videos and outlines the glottic opening, vocal folds, and supraglottic structures through an algorithm based on convolutional neural networks. The segmentation process consists of two phases.

Initially, the RoI is identified by a bounding box. This is accomplished using a CNN, and the result is a low-resolution heatmap that indicates the positioning of the larynx in the image. The processing speed is optimized by identifying the RoI, and redundant convolution operations in areas without the target are reduced. Figure 3 shows the architecture of the fully convolutional network that was used to produce the heatmap. It inputs a laryngeal

image of size 256 × 192 and outputs an image of size 16 × 12. The hidden layer comprises five layers, each featuring convolutional blocks, batch normalization, scaling, and ReLU activation. The initial four blocks are followed by max pooling to reduce the dimensions. A sigmoid function wraps up the network, ensuring an output value between 0 and 1.

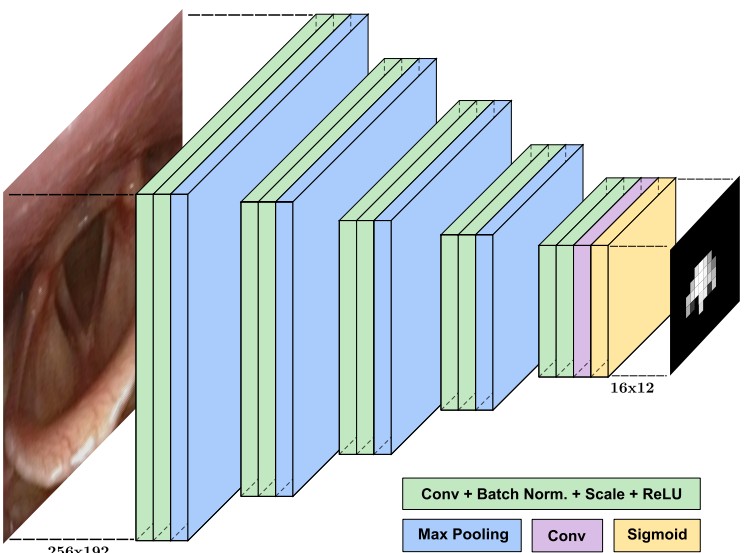

**Figure 3.** Convolutional neural network that outputs a heatmap based on a laryngeal image [12].

In the second phase, segmentation was performed on the identified RoI of the image, treating it as a six-class pixel-wise classification task. This included categories such as the background, glottic opening, vocal folds, and supraglottic structures. An FCN model (Figure 4) with an encoder–decoder design was used for end-to-end pixel label predictions. This model had two main phases: the contractive phase for downsampling and feature extraction, and the expansive phase for upscaling and integrating spatial data. The model processed 256 × 256 RGB images and produced six-channel grey-level images of the same size, with each channel representing predictions for the respective classes.

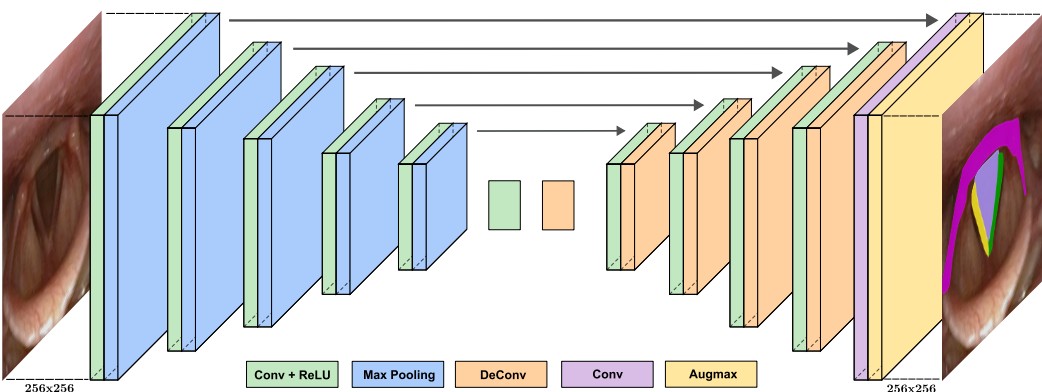

**Figure 4.** Fully convolutional network for pixel-wise segmentation of the larynx [12].

The researchers clearly indicate that this approach ensures efficient and reliable processing of laryngoscopic video footage, achieving a processing speed of eight frames per second. The proposed algorithm calculates the angle between the vocal folds in addition to a measure of the supraglottic movement. As for future improvements, the authors suggest an improved "end-to-end training and prediction without sacrificing its efficiency". In the future, the training process will improve with a bigger and more varied set of data, which will include patients with diverse laryngeal conditions. Adding a large number of normal laryngeal scans, taken both when patients are at rest and exercising, will make the existing

data even better. This will help set standard ranges for the signals being studied, making it easier for doctors to understand the results.

### 4.2. Mask R-CNN

In the study by Choi et al., the Mask R-CNN architecture demonstrated promising results for the segmentation of anatomical structures [17]. While the model is primarily designed for ETI during emergencies, its training in vocal fold and epiglottis segmentation makes it adaptable to the field of pure larynx segmentation.

The model was trained using the Mask R-CNN architecture [18]. Mask R-CNN, an extension of Faster R-CNN [26], serves as an instance segmentation technique, offering both object detection and mask segmentation capabilities. Figure 5 shows the architectural design of the network. The input image is fed into a backbone network, specifically a ResNet Feature Pyramid Network (ResNet-FPN) for feature extraction. The extracted features are input to a Region Proposal Network (RPN). The RPN is a fully convolutional network that predicts object boundaries, suggesting potential bounding boxes around objects in the image. However, variations in bounding box alignment arise due to quantization and pooling operations. To address this, a RoIAlign operation is applied to each candidate bounding box, ensuring proper alignment of extracted features with the input image. Subsequently, fully connected layers predict the object class in addition to applying bounding box regression for each extracted candidate box. Simultaneously, a fully convolutional network (FCN) predicts a mask for each detected object from the candidate boxes. The whole method results in three outputs for each object: a bounding box, class, and segmentation mask.

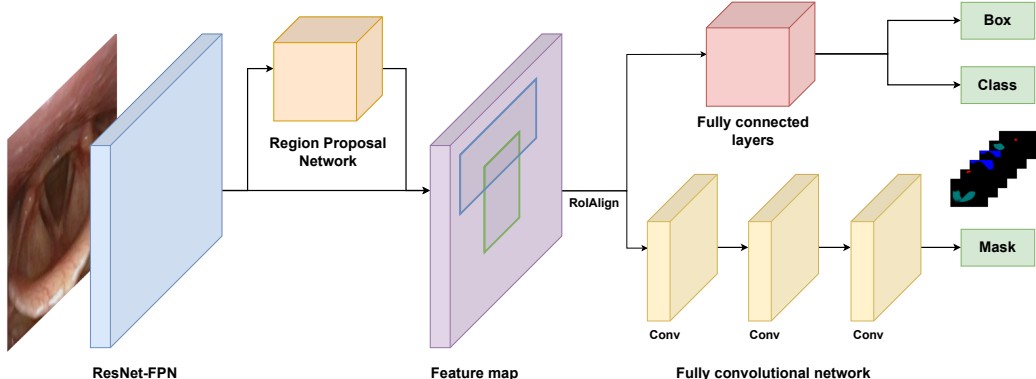

**Figure 5.** Mask R-CNN architecture for object detection, classification, and segmentation [18].

Compared to multi-class FCN, Mask R-CNN decouples the segmentation and classification process. Therefore, multiple masks with the same class may occur. Choi et al. configured the model with a refinement process, dropping masks with low-class prediction scores and aggregating instance masks with the same class.

As for the evaluation, the Mask R-CNN performed well, particularly in accurately identifying the epiglottis and vocal fold structures. These achievements make the model an interesting asset for future research on image segmentation related to EILO. Further, the Mask R-CNN achieved an inference time of 32 frames per second—the best result amongst the tested models from the study. To enhance the performance in the future, the authors suggest developing supplementary ML methods for better processing of noisy images.

### 4.3. 3D VOSNet

Compared to the other two methods, the 3D VOSNet contains sequential time-series information [19]. Using the image segmentation of video frames in a model with sequential information allows for a more accurate and context-aware understanding of object behaviour over time. This approach improves tracking reliability, anomaly detection, and overall model performance. The model retains time-series data from three images before and after each image, allowing it to tackle changes in position and obstructions.

This means the model can reliably segment and identify elements in laryngoscopy videos without being affected by external factors like camera shake or occlusion. Besides using the model for laryngeal segmentation, this study also incorporated a laryngeal identification algorithm. This algorithm calculates six key metrics: vocal fold length, vocal fold area, vocal fold curvature, length deviation, glottal area, and vocal fold symmetry. The mentioned metrics help ensure objective and reliable decision-making during diagnostics and aid in clearly explaining laryngeal conditions like vocal fold paralysis to the patient post-diagnostics.

For contextual information, the model includes the last three and next three images for each input image, which allows it to refer to the features of the surrounding images. The neural network, shown in Figure 6, consists of an encoder and a decoder. The seven input images go straight into a convolution layer of size $7 \times 7 \times 7$ for a feature extraction of the sequence. The encoder unit is composed of four convolution blocks and 16 identity blocks. Each convolution block employs an Inception design [27], housing 65 layers of $1 \times 1 \times 1$ convolutions alongside 32 layers of $3 \times 3 \times 3$ convolutions. In contrast, the identity block fuses an Inception framework with a residual element and consists of 64 layers featuring $1 \times 1 \times 1$ convolutions, as well as 32 layers with $3 \times 3 \times 3$ convolutions. The distinguishing feature between the convolution block and the identity blocks is the additional convolutional layer in the shortcut path of the convolution block. In the core architecture, ResNeXt [28] is utilized to merge the Inception framework with ResNet's residual blocks, enabling the extraction of features at multiple scales without the issue of gradient disappearance. In the training process, Dice Loss focuses on evaluating targeted areas, such as vocal folds, to correct imbalances between the object of interest and the background. When combined with Categorical Focal Loss, which puts higher weight on intricate background details, the two-loss metric provides a balanced approach to training the model effectively. The model results in an image segmentation algorithm with great metrics that provide segmentation of the glottic structures.

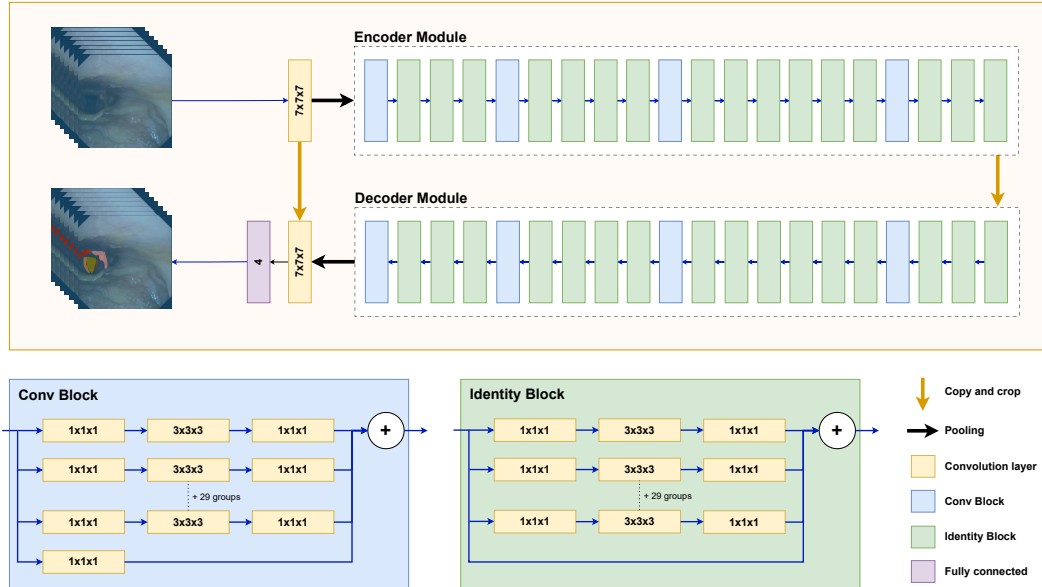

**Figure 6.** 3D VOSNet architecture for segmentation of a sequence of laryngeal images [19].

## 5. Discussion

Machine learning methods for EILO diagnostics use algorithms and data to automatically learn patterns of the laryngeal structures. As an assisting tool for clinicians, these methods can potentially improve the objectivity and consistency of EILO diagnostics by reducing human errors and biases. Our findings also reveal that the integration of machine learning with EILO diagnostics remains a largely untapped research domain, providing significant room for further exploration.

This paper presents three selected ML methods with different approaches, offering great results on laryngoscope image segmentation tasks. Lin et al. trained convolutional networks in a two-step approach, first identifying RoI and then performing segmentation. The RoI identification clearly helps reduce training duration and improve results. The Mask R-CNN implementation used by Choi et al. is based on the same principles, but RoI and segmentation are contained within the same model, making it more complex. The 3D VOSNet's sequential training keeps contextual information on laryngeal movements, thereby increasing the model's performance. The models have their advantages and disadvantages, but they all stand out as interesting choices for further investigation within EILO diagnostics with ML.

A challenge significantly impacting the application of machine learning for EILO diagnostics is the variability in CLE-test data and anatomical differences between individuals. As the video quality varies, the laryngeal structures are not always easy to detect, even for the human eye. Issues with data quality can also arise from factors such as inconsistent visualization of the larynx due to camera placement, camera lens covered by spit, or the epiglottis obscuring view during swallowing motions. These challenges are compounded by the variable audio quality, where certain frequencies may require filtering to ensure clear audio data for accurate analysis. Additionally, differing recording equipment and file formats across clinics make the development of a standardized, universally applicable machine learning solution challenging. Strict health data privacy regulations need to be handled carefully to provide the secure use of patient data. It may inhibit the machine learning training phase and make obtaining enough variety in the training data more difficult. Machine learning methods for EILO diagnostics require a large amount of high-quality data for training and validation. Our research discovered an open-access dataset containing laryngeal images [29]. Initially intended for training models focused on laryngeal diseases, this dataset can be modified for laryngeal structure segmentation through the inclusion of manual labelling. A higher availability of additional open-access data with pre-existing labels would benefit future research in this domain. Moreover, both Choi et al. [17] and Chen et al. [19] may provide their respective datasets on request.

ML methods may also lack transparency or interpretability, meaning that the rationale or logic behind their predictions or classifications may not be clear or understandable to clinicians or patients. Moreover, these methods may not be generalizable, meaning that they may not perform well on different populations, settings, or devices than those used for training or validation. Therefore, machine learning methods for EILO diagnostics should be carefully designed, evaluated, and implemented, considering the clinical needs, ethical implications, and technical challenges.

## 6. Conclusions

The integration of ML techniques for EILO diagnostics has the potential to help clinicians. Computer vision ML methods like image segmentation can delineate laryngeal structures and achieve a more objective assessment. While challenges persist, especially in differences in patients' laryngeal anatomy, the synergy of ML and medical expertise for EILO assessment is an important field to explore in the years to come.

As a final remark, the authors of this paper intend to investigate the segmentation and analysis of laryngeal structures further in an upcoming study, building upon the findings presented in this paper.

**Author Contributions:** Conceptualization, R.M. and R.A.; methodology, R.M.; investigation, R.M.; resources, H.K.K.; writing—original draft preparation, R.M. and R.A.; writing—review and editing, H.K.K., H.C., O.D.R. and R.A.; visualization, R.M.; supervision, R.A.; project administration, R.M. and R.A. All authors have read and agreed to the published version of the manuscript.

**Funding:** This research received no external funding.

**Institutional Review Board Statement:** Not applicable.

**Informed Consent Statement:** Not applicable.

**Data Availability Statement:** No datasets were used or created for this study.

**Conflicts of Interest:** The authors declare no conflicts of interest.

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
