# Peer review of "Diagnostics of Exercise-Induced Laryngeal Obstruction Using Machine Learning: A Narrative Review"

_electronics, doi:10.3390/electronics13101880_

Round 1

Reviewer 1 Report

Comments and Suggestions for Authors

The manuscript explores machine learning methods for exercise-induced laryngeal obstruction (EILO) diagnosis, including Convolutional networks, Mask R-CNN and 3D VOSNet. The manuscript also reveals that the integration of machine learning with EILO diagnostics remains a largely untapped research domain, providing significant room for further exploration. It has important practical application value.

The questions are:

1. About the ";" punctuation in the title of the manuscript, it is better to revise it as ":".  <Diagnostics of Exercise-induced Laryngeal Obstruction using Machine Learning: A narrative review>

2. What are the main contributions of this manuscript ? Or what are the main difference between your review and other review about the ML method for diagnostics of EILO?

3. It is better that if the authors use a table to show the comparison of the experimental results of these machine learning methods for diagnostics of EILO.

4. In the related reserch that has been published, are there any open dataset about the diagnostics of EILO?  As a review, it is better to show the link of the open dataset.

Reviewer 2 Report

Comments and Suggestions for Authors

This paper presents machine learning methods for exercise-induced laryngeal obstruction (EILO) diagnosis.

Comments are given below in order to improve the paper

1.      The discussion of related research works needs to be improved in order to demonstrate the reason for the use of the presented algorithm. The difference of the given research from previous ones should be explained.

2.   The contrıbutıons of the paper should be mentioned in section 1.

3.   The presentation of the paper is not good. The authors need to give flow charts or basic stages of the presented algorithm.

4.      Training and testing division should be demonstrated in the experimental section. Convergence graphics should be provided.

5.   Parameter update formulas should be presented and explained

6.      The results of simulations, and segmentation results should be depicted.

7.      The result of simulations using performance criteria- loss function, accuracy, F1 score, precision, etc. should be presented.

8.      Comparative results with other techniques and state-of-art analysis should be given.

Comments on the Quality of English Language

minor

Round 2

Reviewer 1 Report

Comments and Suggestions for Authors

 Accepted

Author Response

Thank you so much for accepting the paper and for dedicating your time to reviewing it!

Reviewer 2 Report

Comments and Suggestions for Authors

The authors presented a review of existing research on the diagnosis of exercise-induced laryngeal obstruction (EILO). However, the presentation of the article is not very good. Authors are required to provide a table listing research studies related to the diagnosis of EILO. The table should include the methodologies, results obtained, and performance analysis of the methodologies used for EILO diagnosis. The status of the data sets needs to be described. Evaluation and validation metrics need to be presented. Future studies need to be indicated.

Comments on the Quality of English Language

minor

Round 3

Reviewer 2 Report

Comments and Suggestions for Authors

The authors have addressed my comments. The paper can be accepted for publication.